# A BIOLOGICALLY-INSPIRED FOVEATED INTERFACE FOR DEEP VISION MODELS

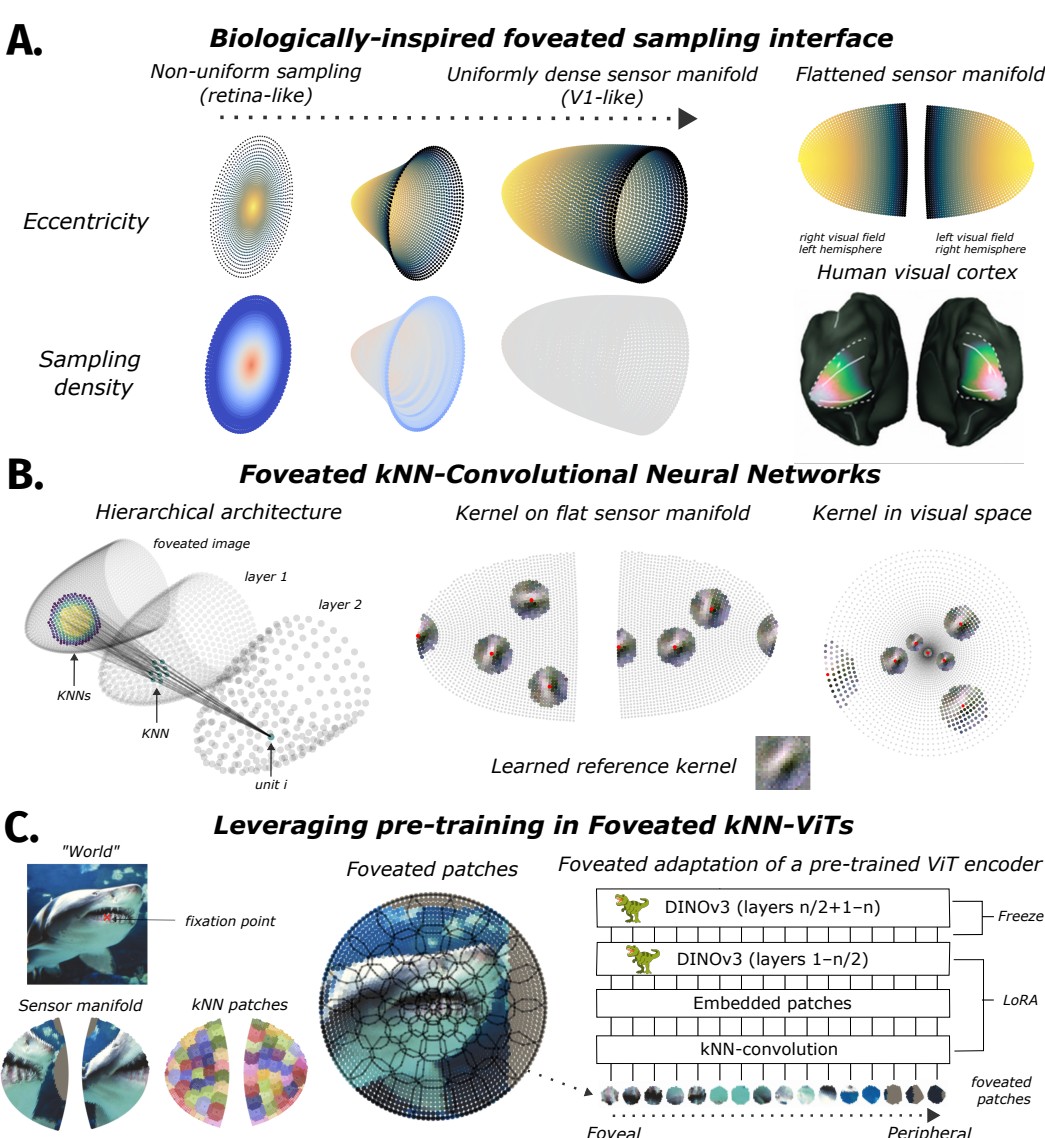

**Figure 1:** A biologically-inspired foveated interface for deep vision models. **A.** Foveated sensing is equivalent to uniform sampling on a magnified sensor manifold (Rovamo & Virsu, 1984). When the manifold is divided along the vertical meridian and flattened, its relationship to the two hemispheres of primary visual cortex is evident. **B.** Building hierarchical convolutional networks via uniform k-nearest-neighbor (kNN) sampling on the sensor manifold. Our kernel mapping algorithm allows us to perform kNN-convolution of a filter across the sensor manifold, yielding eccentricity-dependent receptive field sizes. **C.** Building vision transformers (ViTs) from a kNN-convolution-based patchification of the sensor manifold. Low-rank adaptation of the early layers allows for successful adaptation of off-the-shelf foundation models, enabling high performing foveated ViT models.

## ABSTRACT

Human vision is foveated, with variable resolution peaking at the center of a large field of view; this reflects an efficient trade-off for active sensing, allowing eye-movements to bring different parts of the world into focus with other parts of the world in context. In contrast, most computer vision systems encode the visual world at a uniform resolution over space, raising computational challenges for processing full-field high-resolution image formats efficiently. We propose a biologically-inspired foveated sampling interface that reformats a variable-resolution array of sensors into a uniformly dense, curved sensor manifold. Receptive fields are defined as k-nearest-neighborhoods (kNNs) on the sensor manifold, and we develop a novel kernel mapping technique to enable kNN-convolution. We demonstrate two use cases: (1) a novel kNN-convolutional architecture that natively learns features over foveated input, and (2) an integration of our foveated interface into the vision foundation model DINOv3 via low-rank adaptation (LoRA). These models maintain or improve accuracy compared to non-foveated counterparts, and open pathways for scalable active sensing and efficient modeling of increasingly high-resolution visual data.

## 1 INTRODUCTION

Processing the visual world in its native high resolution poses serious computational challenges. Notably, within deep learning, computer vision has typically simplified the problem by working with low resolution images, with 224x224 being typical. As computer vision advances, processing the native high-resolution of the world, while being sensitive to a large field-of-view, is a critical goal that can enable key applications in robotics, self-driving cars, and broader scene processing. For transformer-based architectures (Dosovitskiy et al., 2020), increasing resolution (i.e. image side length) is a doubly quadratic cost; first, it is quadratic in the number of pixels produced, and second, it is quadratic in the attention operation, where every image patch must attend to every other image patch. Efficient solutions are thus a priority.

Here, we turn to the human visual system for inspiration, as it can process information at very high resolutions near the center of gaze (the fovea), while simultaneously representing a very large field-of-view (~180°) with progressively lower resolution moving farther from the center of gaze. These changes in acuity are attributed to the high density of both cone photoreceptors and retinal ganglion cells (RGCs) in the fovea, and the progressively lower density of both as distance from the fovea increases (Watson, 2014). Information from the retina is mapped to a uniform representation via the thalamus into the primary visual cortex (V1), where more cortical area is dedicating to foveal vs. peripheral regions (known as "cortical magnification"; Daniel & Whitteridge (1961). Thus, not only does the human visual system capture more detailed images at the fovea, but it also devotes more cortical real-estate to processing information at the center of gaze.

Why is the human visual system foveated? Some back-of-the-envelope calculations provide some intuition (details are provided in Figure S2). If we first assume that V1 were to dedicate as much space to the full visual field as it normally does to central vision, this would result in the long-axis of V1 expanding to 1.2 meters – approximately 25x its typical length – catastrophically increasing both space and energy demands. If we instead assume a fixed amount of cortical real-estate with the peak resolution throughout, only a $3°$ field-of-view would be possible, impairing broader visual interactions with the world, such as navigation and detection. Thus, foveated sampling of the visual environment provides a balanced solution to the trade-off between resolution, field-of-view, and spatial efficiency, opting for high-resolution over a limited field-of-view, and lower resolutions over larger fields-of-view, with a modest spatial and energetic cost.

## 2 PRIOR WORK

While foveated sensing is not a dominant approach in computer vision, it has historically received substantial interest (Weiman & Chaikin, 1979; Javier Traver & Bernardino, 2010; Wang et al., 2021; Da Costa et al., 2024; Jérémie et al., 2024). Generally, two forms of foveated sensing have been

explored: a log-polar image model Weiman & Chaikin (1979); Javier Traver & Bernardino (2010); Jérémie et al. (2024), which samples radius (eccentricity) logarithmically and selects an equal number of angular samples at each radius, and warped Cartesian approaches Basu & Licardie (1995); Lukanov et al. (2021); Wang et al. (2021); Da Costa et al. (2024), which re-project log-polar images back into Cartesian space to produce a warped image that over-represents the center of gaze. These approaches share a common issue, derived from their attempt to produce a rectangular grid-like foveated image. The result is locally *anisotropic* sampling (locally, the sampling rate differs with respect to polar angle and radius), a non-biologically-plausible property that produces undesirable warped receptive field shapes (Figures S4, S5). A notable exception is the recent model of Killick et al. (2023); however, their approach requires the use of fixed Gaussian-derivative basis functions in place of learned spatial kernels, and is thus not directly comparable to the other methods which allow for end-to-end learning of both spatial and feature-based representations. Similarly, Cheung et al. (2017) demonstrated the emergence of foveated sampling in a scenario where the sampling grid was learned, but the irregular structure of the learned grid did not support convolutional processing. We discuss these approaches in greater detail in Appendix 8.4. Last, some approaches have foregone a consideration of foveated *sensing*, and have modeled foveated perceptual processing purely at the architectural level, by focusing resources more on central vs. peripheral image content (Kerr et al., 2025; Chuang et al., 2025), showing benefits in robotics applications. While promising, these approaches leverage a discrete set of processing resolutions rather than a continuum, and are not designed with sensing efficiency in mind. A foveated sensor can provide additional efficiency gains, particularly in sim2real pipelines (Pinto & Gupta, 2016), where reducing the numbers of rays traced can reduce both computational and memory-based resources. Overall, prior work has introduced a variety of mechanisms for implementing foveation in computer vision models, but there is not yet a general-purpose implementation that shows well-behaved and biologically-plausible visual field sampling, that can be adopted across diverse architectures.

## 3 SUMMARY OF CONTRIBUTIONS

Here, we make three key contributions.

1. We introduce a new *foveated sampling interface* to deep vision models. Visual space is sampled according to a mathematical model of the retino-cortical mapping (Rovamo & Virsu, 1984), as shown in Figure 1A; our novel k-nearest-neighbor (kNN)-convolution and kernel mapping method enables perceptual processing over this foveated input format. This sampling interface draws on known characterizations of the primate visual system: cutting this manifold on the long-axis – corresponding to the vertical meridian dividing left and right visual fields – yields a strong first-order match to the retinotopic organization of human V1.

2. Second, we present a *novel kNN-convolutional neural network* that natively learns convolutional features over foveated input (Figure 1B). We train these models end-to-end, while varying the degree of foveation, and demonstrate both biologically-plausible spatial receptive field properties, and an advantage for intermediate foveation levels in image classification compared to non-foveated control models.

3. Third, we show how to *outfit a state-of-the-art pre-trained vision transformer with a foveated sensing interface* (Figure 1C). Our method uses the kNN-convolution to implement a foveated patch embedding into an otherwise standard vision transformer, and we explore fine-tuning protocols leveraging low-rank adaptation (LoRA) (Hu et al., 2021) to functionally integrate the new sensor into the DINOv3 model. This model achieves high performance at a reduced computational budget, beating a matched non-foveated variant with the same limited resource constraints, while unlocking possibilities for efficient active sampling in high resolution settings.

## 4 A BIOLOGICALLY-INSPIRED FOVEATED INTERFACE

In standard computer vision models, the representation of the visual world is rectangular (the size of the image), and the model can perform regular (i.e. convolutional) processing directly over the input image using rectangular kernels or patches (e.g. 3x3 pixel) that evenly tile the activation map. This

regular processing relies on the presence of a uniformly dense representation of the image, here, a regular grid. However, we seek to sample points in a foveated manner where resolution depends only on eccentricity – but not polar angle; this set of points can be reformatted into a uniformly dense, but curved, manifold (Figure 1A). Thus, we introduce a foveated interface with two components. First, we follow a mathematical model of the retino-cortical mapping introduced by Rovamo & Virsu (1984) (discussed further in Appendix 8.1 and Figure S1E), that generates a foveated sensor array and supports an equivalent uniformly dense V1-like representation, which we term the *sensor manifold*. Second, we introduce a kNN-convolution operation, supported by a novel kernel mapping technique, to support perceptual processing on the sensor manifold.

## 4.1 FOVEATED SAMPLING WITH A UNIFORMLY-DENSE SENSOR MANIFOLD

To achieve an *even sampling of points on the 3D sensor manifold*, we begin by sampling points locally isotropically in Cartesian visual space according to the cortical magnification function (CMF); given the CMF $M(r) = \frac{1}{r+a}$ (see Figure S1A), we sample a range of radius (eccentricity) values from the fovea to the periphery equally in the logarithmic dimension given by the CMF integral ($w = \log(r + a)$; see Figure S1B,C). Second, we determine the number of equally spaced angular samples to draw in a circle at each radius in order to preserve local isotropy; that is, to ensure that the distance between neighboring angles is equal to the distance between neighboring radii at any given point. This ensures locally consistent spacing (local isotropy) throughout the visual field, while achieving magnification along the radial dimension (Figure S1D). Together, this sampling strategy produces points that are approximately evenly distributed on the 3D sensor manifold. The "complex log" model of Schwartz (1980) can be used as an ideal "flat" representation that allows for 2D visualization of the entire sensor manifold–a cut is made along the vertical meridian, and the two hemifields are flattened, mirroring the spatial organization of visual information in the hemispheres of area V1 (Figure 1A, right; Figure S1E).

To control the *degree of foveation* in our models, we vary the $a$ parameter in the CMF, corresponding to the size of the fovea, or area of highest resolution (Schwartz, 1980): $M(r) = \frac{1}{r+a}$. A small fovea (small $a$) is indicative of strong foveation, whereas a proportionally large fovea (large $a$) corresponds to weak foveation, approaching uniform sampling as $a \rightarrow \infty$ (Figure 3A). As shown in Figure S1A-C, a desired number of visual sampling radii are generated from equally spaced samples of the integrated CMF (or log radius, $\log(r + a)$), from the minimum to maximum value. For a given $a$, we search over the number of radii $n_r$ that produces the closest match to a target number of samples $n$, ensuring approximately matched resources across models (see Appendix 8.1 for further detail).

## 4.2 KERNEL MAPPING FOR kNN-CONVOLUTION ON THE SENSOR MANIFOLD

To perform perceptual processing on the sensor manifold, we specify spatial receptive fields as k-nearest-neighborhoods (kNNs) around a set of output units tiled across the same manifold. To enable convolutional weight-sharing across kNNs on the manifold, we introduce the **kNN-convolution**, mediated by a novel **kernel mapping** technique, which maps a reference kernel ($W$) – learned in a standard Cartesian grid – into each neighborhood (Figure 2). A goal of this mapping is to achieve convolutional kernels that are aligned across locations in visual space, such that a vertical orientation filter detects vertical features across the entire image. To do so, we define a set of polar neighborhood coordinates for each kNN, setting the radius $r$ as the geodesic distance from the output unit on the sensor manifold, and the polar angle $\theta$ as the polar angle in *visual* space, computed with respect to the output unit (Figure 2, step 1). We then determine the Cartesian neighborhood coordinates using the standard formulas $x = r\cos(\theta)$, $y = r\sin(\theta)$, achieving aligned visual angles (Figure 2, step 2); these coordinates lie in the same frame as the learned reference kernels ($W$). Finally, we spatially sample the reference kernel with each neighborhood using the reference frame coordinates, shown in Figure 2, step 3. Plotting kernels back into the flattened sensor space highlights their uniform size on the sensor manifold; projecting the learned feature kernel back into visual space shows how the learned filter is the same across visual space, with different visual field coverage depending on eccentricity.

There is an additional choice of the native resolution of the reference kernel. For example, this can be set at the same resolution of the kNNs (square side length $s = \sqrt{k}$), or at a higher-resolution to better accommodate idiosyncratic spatial positions across kNNs (c.f. anti-aliasing). In practice,

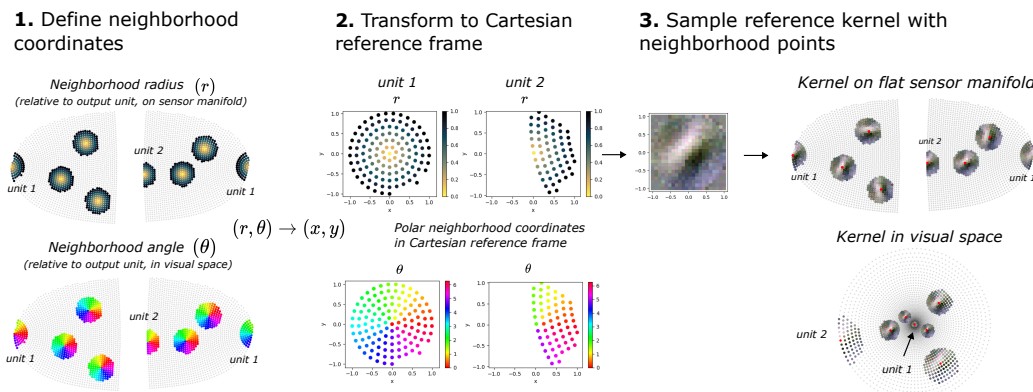

**1.** Define neighborhood coordinates

**2.** Transform to Cartesian reference frame

**3.** Sample reference kernel with neighborhood points

*Neighborhood radius $(r)$*
*(relative to output unit, on sensor manifold)*

*Neighborhood angle $(\theta)$*
*(relative to output unit, in visual space)*

unit 1  unit 2

*Kernel on flat sensor manifold*

$(r, \theta) \rightarrow (x, y)$

*Polar neighborhood coordinates in Cartesian reference frame*

*Kernel in visual space*

**Figure 2:** Kernel mapping procedure for the kNN-convolution operation. First, even k-sized neighborhoods are defined in the sensor manifold (upper), along with their orientation in the visual input (cartesian) space (lower). Second, kernels are transformed into a common cartesian reference frame, and aligned to a common reference kernel. Third, we visualize the learned reference kernel across different units in both the flattened representational space (top), and in visual space (bottom). These show how the kernel is a fixed size in the sensor manifold, and orientionally-aligned in visual cartesian coordinates, while scaling with eccentricity.

we find that the latter choice leads to stronger performance (Figure S10). We make this our default, using a square kernel of side length $s = 2\sqrt{k}$. Notably, while this does increase parameter count, it is heavily constrained by the fixed spatial mapping, and each mapped kernel still uses the same number of parameters as a standard 2D kernel of side length $s = \sqrt{k}$.

## 5 FOVEATED KNN-CNNS

Given this sensor interface, we next built a foveated convolutional architecture, which leverages multiple layers of kNN-convolution operations, supporting hierarchical foveated feature learning. Each layer's activation map is formatted as a sensor manifold, with a particular resolution of evenly spaced samples. The resolution of each layer decreases progressively through the network, as in typical CNNs. Each layer defines the kNN centers for processing of the previous layer (Figure 1B).

To instantiate a particular foveated model architecture, we paralleled the choices of a simple AlexNet-like (Krizhevsky et al., 2012) convolutional model, with 5 convolutional layers, 3 pooling layers, and 2 fully connected layers. The first convolutional layer uses a kernel size of 11 and stride of 4, the second convolutional layer uses a kernel size of 5 and stride of 1, and the remaining convolutional layers use a kernel size of 3 and stride of 1. Besides the initial sampling layer, downsampling is performed strictly in (3x3) max pooling layers with stride 2, following the first and fourth convolutional layers. Before the first fully connected layer, a global average pooling layer is used, as in ResNets and other later architectures (He et al., 2016). Across convolutional layers, there are 96, 256, 384, 384, and 256 channels. We implement padding by extending the sampling outside the processing field-of-view, labeling such units as padding units whose activation always maps to 0. In the formation of kNNs in the following layer, these padding units are then automatically selected as nearest neighbors to appropriately pad the input; "unit 2" in Figure 2 is an example unit whose receptive field is padded. Finally, we add two fully-connected layers with 1024 units, use a ReLU nonlinearity, and perform batch normalization after each nonlinearity, with a learned affine transformation (Ioffe & Szegedy, 2015).

### 5.1 FOVEATED CNNS MATCH THE SPATIAL CHARACTERISTICS OF PRIMATE NEURAL RECEPTIVE FIELDS

First, we demonstrate that this model architecture produces a strong match to the spatial characteristics of primate neural receptive fields. In particular, receptive field mapping studies have consistently demonstrated that primate receptive fields are larger the more peripheral they are, and the farther up

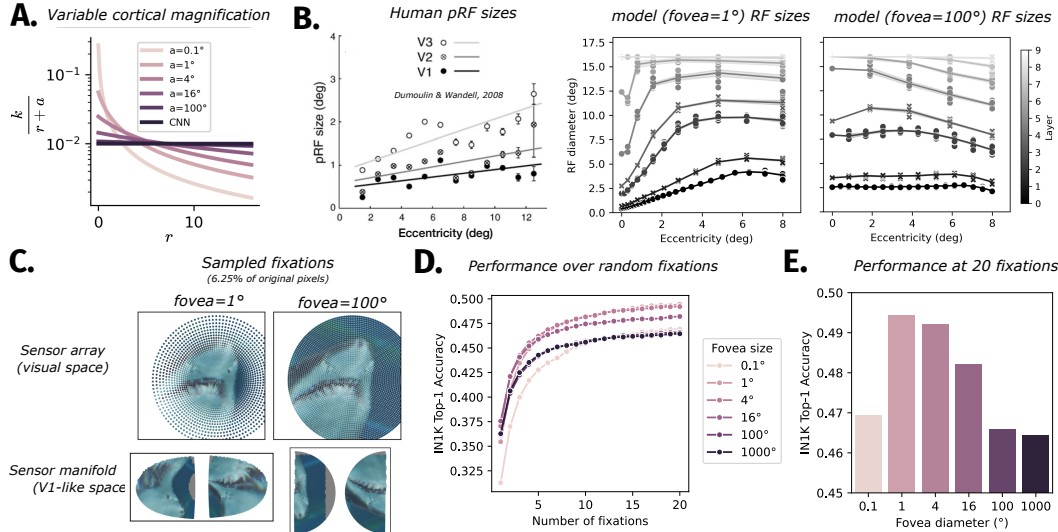

**Figure 3:** Foveated kNN-CNNs account for the spatial characteristics of primate neural receptive fields, and provide a performance advantage in ImageNet classification. **A.** Implementing variable foveation via variability in the cortical magnification function (CMF). Given the CMF $\frac{1}{r+a}$, we can specify a continuum of foveation, where small $a$ corresponds to strong foveation, and uniform sampling is achieved as $a \to \infty$. **B.** Left: human population receptive field (pRF) sizes, measured with fMRI (Dumoulin & Wandell, 2008). Middle: receptive field (RF) sizes across layers in a foveated model ($a = 1$). Right: RF sizes in a nearly uniformly sampling model ($a = 100$). **C.** Sampled fixations for strong and weak foveation models, shown both as a sensor array in visual space, and a (flat) sensor manifold in a V1-like space. **D.** ImageNet-1K results after training, mapping out performance up to 20 random fixations. **E.** Highlight of performance across foveation levels at the maximum of 20 fixations.

the hierachy they are (Dumoulin & Wandell, 2008; Motter, 2009). Data from V1-V3 in humans in Figure 3A depict this phenomenon, showing a linear increase in "population receptive field size" (pRF) size with eccentricity, with an increase in both slope and intercept across hierarchical areas.

We characterized the spatial RF properties in each layer of our model, assessing both a foveated ($a = 1$) and non-foveated ($a = 100$) variant. For each layer, we compute the RF diameter of every feature map location by back-projecting receptive field neighborhoods through each preceding layer until reaching the input layer. We then plot the RF diameter as a function of eccentricity for each layer. As shown in Figure 3B, middle, we see an approximately linear increase in RF size with eccentricity. This linear increase eventually plateaus for each layer once the receptive field centers get far enough into the periphery to incorporate padding units beyond the field-of-view, such that the RF does not continue to grow. Additionally, both the slope and intercept of RF diameter by eccentricity functions increase with the hierarchical layer number. The dependence of RF size on eccentricity, but not hierarchical layer, falls directly out of the foveated sampling, as it is abolished when assessing the non-foveated variant, shown in Figure 3B, right. Further intuition for the eccentricity-dependent size increase can be seen in Figure 2(right), showing example RFs in the first layer on both the flat sensor manifold and visual space. In addition to explaining the qualitative patterns of RF sizes across visual areas, foveated sampling allows us to account for the precise shapes of primate receptive fields; we discuss this in Appendix 8.3.

## 5.2 FOVEATION IMPROVES kNN-CNN CLASSIFICATION ACCURACY

Next, we examine the consequences of foveation for perceptual performance in image classification. Recall that the motivation behind foveation is that it is useful when operating over a much higher resolution input than allowed by the sensor, as in the ambient light field in the real world. Here, we simulated this scenario by giving kNN-CNNs a constrained "pixel-budget" (64x64) to sample from a 256x256 "ambient" resolution, representing a 16-fold reduction in samples. Examples of foveated

samples can be seen in Figure 3C. (Note that since these models were constrained to only sample $64^2$ pixels, we replaced the stride of 4 in the first layer with a stride of 2, in order to prevent feature map resolution from shrinking to 1 before the final convolutional layer.)

We perform experiments using the ImageNet dataset (Russakovsky et al., 2015). For more detailed experimentation across hyperparameter variations, we also use a smaller custom ImageNet subset which we refer to as ImageNet-100. This dataset consists of 100 random ImageNet categories, and contains the same number of images as CIFAR-100 (500 training images per category, and 100 validation images per category); basic validations in result trends across datasets are shown in Figure S6. To facilitate faster training, both datasets are re-sampled to a uniform maximum resolution of 256 for use with FFCV (Leclerc et al., 2023), unless otherwise mentioned. We train our models using 4 random fixations drawn from a central area of the image; in our main experiments we use a radius of 0.25 of the image size to define the fixation zone, however we also experiment with a larger radius of 0.45 (Figure S9). Due to gradient accumulation, it is expensive to train on many fixations, imposing a similar cost (and benefit in learning) to increasing the number of epochs. However, as inference is computationally cheaper and thus more amenable to large numbers of fixations, we allow for up to 20 random fixations during validation. We forego crops, as cropping is a form of foveation that focuses perception on a restricted part of the image at higher resolution than would otherwise be allowed; we examine this choice in Figure S8. Models performed a simple aggregation of information across fixations by averaging the logits of each fixation. During training, the average logits are used with the cross-entropy loss for supervised learning; during validation, these average logits are used to generate top-1 and top-5 accuracies.

Plotting performance over the number of fixations (Figure 3D), we see that each model improves significantly with increasing fixations, generally saturating in improvement by about 20 fixations. Focusing in on the performance at the maximum 20 fixations (Figure 3E), we see an inverted U-shaped function over the fovea size, with peak performance at an intermediate foveation level of $a = 1$, beating out both more strongly foveated ($a = 0.1$) and more uniformly foveated models ($a = 1000$). We discuss these results in further detail, along with a series of follow-up analyses, in Appendix 8.5; briefly, it seems this degree of foveation is well suited to capture the center-bias and scale-bias of the critical object content in these ImageNet images. These results demonstrate the successful application of our foveated sensor and kNN-convolution, highlighting an example performance benefit for foveation under resource constraints.

## 6 FOVEATED kNN VISION TRANSFORMERS

Last, we demonstrate that it is possible to apply a foveated interface to a vision transformer (ViT), and offer a recipe for converting an off-the-shelf foundation model into a foveated variant using low-rank adaptation (LoRA; Hu et al., 2021, see Figure 1C).

### 6.1 PATCHIFICATION THROUGH kNN-CONVOLUTION

The key to connecting the foveated interface to a transformer architecture is developing a suitable patchification scheme. We leverage our kNN-convolution. Precisely, we define two foveated sampling grids: 1) a sensor array, and 2) a patch-center array. Patches are defined as kNNs over the sensor array, using distances on the sensor manifold as typical in the kNN-convolution. We choose the length of the patch-center array to exactly match the number of patches in a baseline ViT, by constraining the set of $a$ values to those that produce the desired number of patches; this procedure is explained further in Appendix (8.8). For $n = 64$ patches, we determine 5 suitable $a$ values (rounded here to two decimal places): 0.03, 0.17, 0.82, 4.61, 115.63. We set the patch size $k$ to the minimum such that all sensor locations are included in at least one patch-center kNN; due to the circular nature of kNNs, this induces a small degree of overlap, similar to the use of overlapping convolutions in the Swin ViT (Liu et al., 2021). An example patchification scheme is shown in Figure 1C.

Broadly, this interface allows ViT architectures to process the image in a foveated manner, with more tokens dedicated to the center of gaze and progressively less dedicated to the periphery, dependent on the magnification parameter $a$. Compared to CNNs, the ViT architecture only requires a single kNN-convolution for patch embedding, rather than one per layer, which allows for increased efficiency, since the kNN-convolution introduces some overhead relative to 2D convolutions, overhead which

is very minimal for a single convolution with a small feature map size, as in the patch embedding. If the ViT is pre-trained, given the kNNs defined by the patchification, it is also possible to make use of pre-trained patch embeddings as the reference kernel in kNN-convolution, which is then mapped into the patches using the standard kernel mapping procedure (Figure 2).

## 6.2 ADAPTING A PRE-TRAINED ViT FOR FOVEATED PROCESSING

We next explore converting DINOv3 ViT-S(16) (Siméoni et al., 2025) into a foveated variant. In pilot experiments, we determined a suitable strategy for adapting DINOv3 to work with foveated inputs. This strategy uses LoRA (Hu et al., 2021) over the first half of the network, which significantly reduces overfitting relative to full fine-tuning of the network, better retaining the visual feature representations acquired during pre-training while adapting to the foveated inputs. We describe our process for determining this strategy, along with tuning analyses of the hyperparameter $a$, in Appendix 8.7. We select $a = 4.61$, which achieves moderate foveation (Figure 1C).

We compare our model to three baselines. The first is a matched **NonFov-KNN baseline**, using $a = 115.6$ to minimize foveation, while retaining the same patch embedding method. The second is a **uniform baseline**, where the image has been downsampled to 64x64 to match the number of pixels sampled by the foveated variant, and kernels are similarly downsampled from 16x16 to 8x8. This allows the models to be closely matched in total GFLOPs, besides a small overhead of the kernel mapping in KNN-convolution (Table 1). We find that applying the same LoRA finetuning strategy allows this network to improve its performance modestly above its off-the-shelf performance, allowing the network to adapt to the lower resolution (Figure S14. Additionally, we test a matched **log-polar baseline**, in which images are sampled according to the Log-Polar foveation model (Javier Traver & Bernardino, 2010) at the desired resolution, and then are otherwise processed identically to the **uniform baseline**. Using IN-100, we tune the $a$ parameter and select the best performing model. Similarly, we find that the LoRA finetuning strategy improves performance relative to frozen weights, and thus apply it identically to the other models (Figure S14. Models are trained using 4 random fixations, and evaluated using 20, using an identical protocol to that used for CNNs in the previous section. Results are reported in Table 1. We find that our foveated variant significantly beats all three baselines, setting what is to our knowledge state-of-the-art performance on IN-1K at a resolution of $n \leq 4096$ pixels. This indicates that our method for foveation can improve performance in resource-constrained ViTs.

| Model | IN-1K top-1 | # Pixels/fix | # Patches/fix | GFLOPs/fix |
|---|---|---|---|---|
| **Fov-kNN** ($a = 4.61$) | 0.741 | 4025 | 64 | 6.20 |
| NonFov-kNN ($a = 115.6$) | 0.724 | 4025 | 64 | 6.20 |
| Uniform baseline | 0.717 | 4096 | 64 | 6.11 |
| Log-polar baseline ($a = 4$) | 0.689 | 4096 | 64 | 6.11 |

**Table 1:** Performance comparison of different resource-constrained DINOv3 variants ("fix" is shorthand for fixation), and GFLOPs are reported per image.

## 6.3 EFFICIENT SENSING AND PERCEPTION

Last, due to the sparsity of suitable high-resolution benchmarks, we demonstrate the possible efficiency gains of foveated ViTs by performing an analysis of compute requirements for a hypothetical scenario of processing high resolution images. First, we analyze the GFLOPs/image in DINOv3 (ViT-S(16)), breaking down the GFLOPs into attention-based processing, and non-attention-based processing (Figure 4A). We define the image resolution in terms of the side length of a Cartesian image ($m = \sqrt{n}$, where $n$ is the number of samples taken in a resource-matched foveated model, as researchers usually specify $m$ rather than $n$. We find that, for $m < 400$, the cost of non-attention operations (i.e. linear layers, nonlinearities, normalizations, etc.) outweighs the cost of attention-based operations. However, for larger resolutions, the attention-based cost becomes enormous. We perform an empirical big-O analysis by fitting a power law to each subset of FLOPs. Since image resolution scales as $O(m^2)$, the non-attention (mostly linear) operations are approximately quadratic ($O(m^{1.76})$), whereas the attention-based operations scale exactly with $O(m^4)$ due to their quadratic dependence on sequence length ($O(n^2)$). This demonstrates the difficulty of scaling transformers to high-resolution inputs.

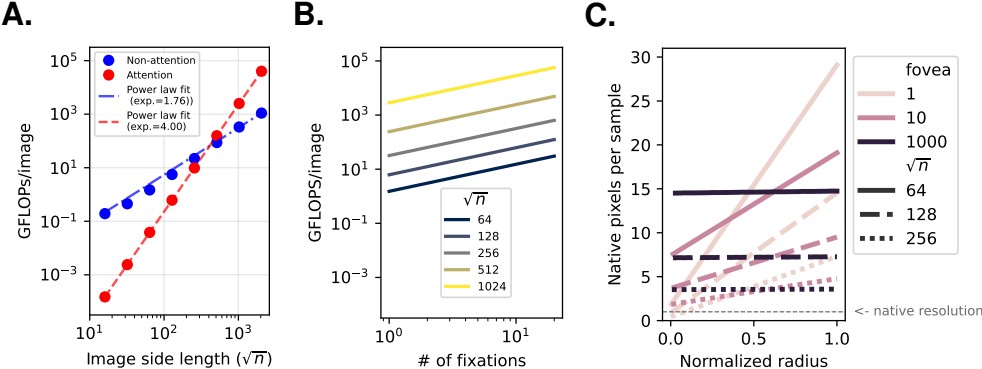

**Figure 4:** Analyzing efficiency in ViT-S(16) from the lens of foveation. **A.** GFLOPs/image as a function of image resolution, separately for attention and non-attention operations. Power laws are fit to each curve as empirical $O(m)$ analyses, where $m = \sqrt{n}$ is the pixels per side of a square image. **B.** GFLOPs/image as a function of the number of fixations, for different image resolutions. **C.** Local sampling resolution (native pixels per sensor sample) as a function of eccentricity, varying the resolution and foveation of the sensor. Reminder: a larger fovea corresponds to more weaker foveation uniform processing. A horizontal line at 1 indicates sampling at the native resolution.

Next, we analyze the GFLOPs/image required to process $n$ sensor locations, setting $m = \sqrt{n}$ to 64, 128, 256. We envision a scenario where the native image resolution is $\sqrt{n} = 1024$, and we can use foveation to achieve higher resolution in parts of the image, while retaining a much broader context than would be afforded by uniform cropping. We compute GFLOPs/image over fixations for each of these sensor resolutions, shown in Figure 4B, additionally plotting $\sqrt{n}$ of 512 and 1024 for reference. We can see that even with many fixations, the lower resolutions are able to reduce computational cost relative to processing a single full resolution image.

Finally, we plot the local sampling resolution (in native pixels per sample) as a function of eccentricity (normalized radius) (Figure 4C). We see that, for strong foveation (smaller fovea parameter $a$), sampling in the fovea can reach the native sampling resolution even at small $n$, but comes at the cost of lower sampling resolution in the periphery. Overall, these plots illustrate the trade-off between the degree of foveation, the number of fixations, the sensor resolution, and the overall compute requirements; this trade-off can be optimized on a case-by-case basis. Notably, strong foveation allows models to achieve a high peak resolution at a lower sampling resolution, while keeping the same large field-of-view and computational demands as models with weaker foveation.

## 7 CONCLUSION

This work presents a novel foveated interface for deep vision models. Our locally isotropic sensor is highly biologically-plausible, with hierarchical convolutional processing producing receptive fields that mimic the spatial properties of primate visual areas V1-V4 (Motter, 2009). With our foveated interface, we unlock a flexible pipeline for exploring resource constraints in deep vision models, by controlling the degree of foveation, with uniform sampling being a special case ($a \to \infty$). Moreover, our foveated models hold strong promise for computational modeling of human vision, for both scientific and technical (e.g. AR/VR) applications.

Notably, the work here is just a start in exploring foveated deep computer vision. Foveation is expected to shine even more in scenarios where higher resolution processing is demanded (Shi et al., 2025), such as large field-of-view naturalistic scenes, or interaction with artificial or natural high-resolution worlds. Moreover, restricting the number of samples can have an added efficiency component in sim2real pipelines in which the foveated sensor can be used to directly control ray-tracing, and thereby limit the required compute for rendering high-resolution scenes. To achieve human-like visual efficiency via foveation, advances are needed both in mechanisms for active vision, as well as saccadic integration – mechanisms that can be built upon our foveated interface in future work.

## USE OF LARGE LANGUAGE MODELS

We disclose use of Large Language Models (LLMs) in this work. LLMs were primarily used in the research ideation process, and in the writing of code for the project, in a collaborative spirit with the authors. LLMs were not used for writing of passages of this paper, and were used sparingly for editing.

## REPRODUCIBILITY

Code will be released on GitHub upon publication.

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

## 8 Appendix

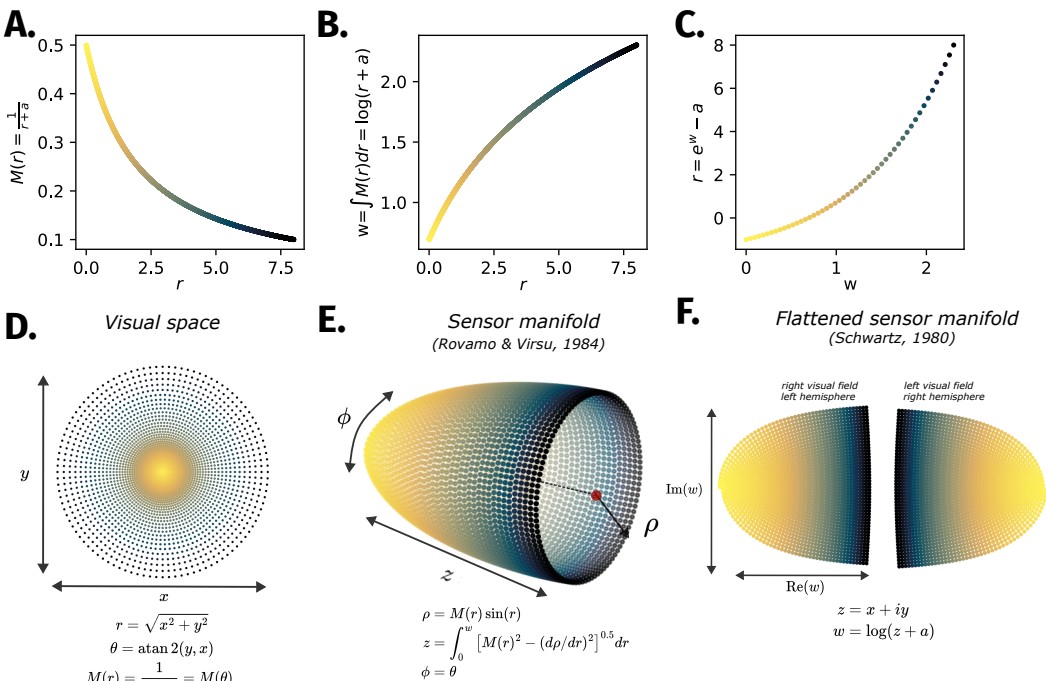

**Figure S1:** The relationship between cortical magnification and isotropic foveated sensing. **A.** The cortical magnification function commonly used to account for the organization of retinotopic maps in visual cortex (Van Essen et al., 1984; Schwartz, 1994). We set $a = 2$ and a field-of-view of 16 degrees. **B.** The integral of the CMF $w$, from 0 to $r$, yielding the "cortical" dimension corresponding to eccentricity. **C.** Sampling evenly along the domain of $w$ and solving for the corresponding retinal radius $r$ to achieve foveated samples in visual space. **D.** Sensor locations in visual space arising from isotropic foveated sampling. Given the radiall samples from $C.$, the number of angular samples at each radius is chosen to approximately satisfy local isotropy (see main text). **E.** Visual points from **D.** mapped into the complex log model (Schwartz, 1980), which is a flat (2D), locally-isotropic representation of visual space. Due to its meridional anisotropy, we can see that sampling is less dense on the top and bottom of the maps, corresponding to the vertical meridian where cortical magnification is maximally different from the horizontal meridian. **F.** Visual points from **D.** mapped in the globally isotropic manifold of Rovamo & Virsu (1984). Due to its global isotropy, points are approximately evenly spaced globally across the manifold.

### 8.1 Detailed explanation of foveated sensor manifold

Formally, in visual space, we consider a polar coordinate system $(r, \theta)$ defined by eccentricity $r$, and meridional angle $\theta$ which is directly sampled by the retina; we refer to the samples in this space as the "sensor array". We then consider a mapping into the primary visual cortical area (V1), what we refer to as the "sensor manifold", defined in cylindrical coordinates $(\rho, z, \phi)$: $z$ corresponding to the axis of rotation (in mm), $\phi$ corresponding to the angle of rotation (in radians), and $\rho$ indicating the distance of the cortical surface from the $z$ axis (in mm). Rovamo & Virsu (1984) developed the following equations for this mapping:

$$\rho = M(r)\sin(w); \quad z = \int_0^w \left[ M(r)^2 - (d\rho/dr)^2 \right]^{0.5} dr; \quad \phi = \theta$$

### 8.1.1 Determining the sampling resolution

Typically, there is a desired number of samples to be made; for the sensor, we desire to match to a target image resolution ($n = h * w$), whereas for convolutional layers, we desire to match to a

target feature map resolution, similarly ($n = h * w$). Our sampling works by first selecting a number of radii; the number of samples is then determined by selecting the number of angles in order to preserve local isotropy at each unique radius. This means we do not have full control over the exact number of samples. One option is to randomly introduce additional angular samples throughout the sampling grid, introducing a mild degree of anisotropy in order to achieve a perfect match in number of samples. The other option is to select the number of radii that produces the closest match to the desired number of sampling points. In practice, we found that the latter option worked better in spite of having less parameters; given that it also produces more isotropic sampling, we consider this the preferable option. Thus, our sensor and kNN-convolution layers always have at most the same number of units as the corresponding image or CNN layer, respectively, and sometimes less. The result of this can also be viewed in terms of equal surface area and sampling density on the sensor manifold, despite different shapes (Figure 3C).

## 8.2 Trade-offs in resolution, field-of-view, and processing resources

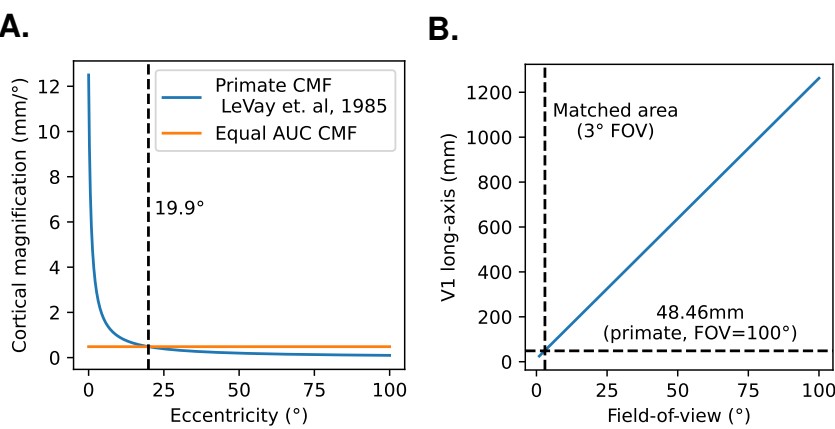

**Figure S2:** Illustration of trade-offs inherent to foveated vision. **A.** Cortical magnification function (CMF) from LeVay et al. (1985), used in our calculations, along with an equal area-under-the-curve uniform CMF. The uniform magnification is comparable to $19.9°$ in the standard CMF, thus affording intermediate-peripheral vision across the visual field for equal V1 size. **B.** Assuming uniform cortical magnification equivalent to the central value (12.5 mm/deg), we solve for the length of V1 by integration, and indicate lines corresponding to matched V1 area and the allowable field-of-view under this uniform magnification.

## 8.3 Accounting for the shapes of primate neural receptive fields

In the main text, we demonstrate that our foveated kNN-CNN accounts for the progressive increase in RF size with eccentricity in primate neural receptive fields. Here, we demonstrate that it also accounts for their shapes. We plot the data of Motter (2009) in Figure S3D. On the left, we see a contour plot of a V4 neuron's visual responses, showing the characteristic non-Gaussian shape, with an elongation of contours along the radial dimension. In the middle, this neuron's response profile is re-plotted as contours on the V1-like surface, demonstrating that the visual response pattern arises from approximately circular (or isotropic Gaussian) sampling on the V1-like surface. On the right, the isotropy is plotted by characterizing the ratio of short vs. long axis ratios in a bivariate anisotropic Gaussian fit over all measured neurons. The density of this distribution is heavily concentrated on a ratio of 1, demonstrating approximate isotropy with no systematic deviations from it. We refer the reader to Motter (2009) for further detail. In Figure S3E., we replicate these analyses in our foveated ($a = 1$) model. Given that our model is based on hierarchical isotropic sampling on the V1-like sensor manifold, it is no surprise that the receptive field properties follow the same trend as the macaque V4 neurons.

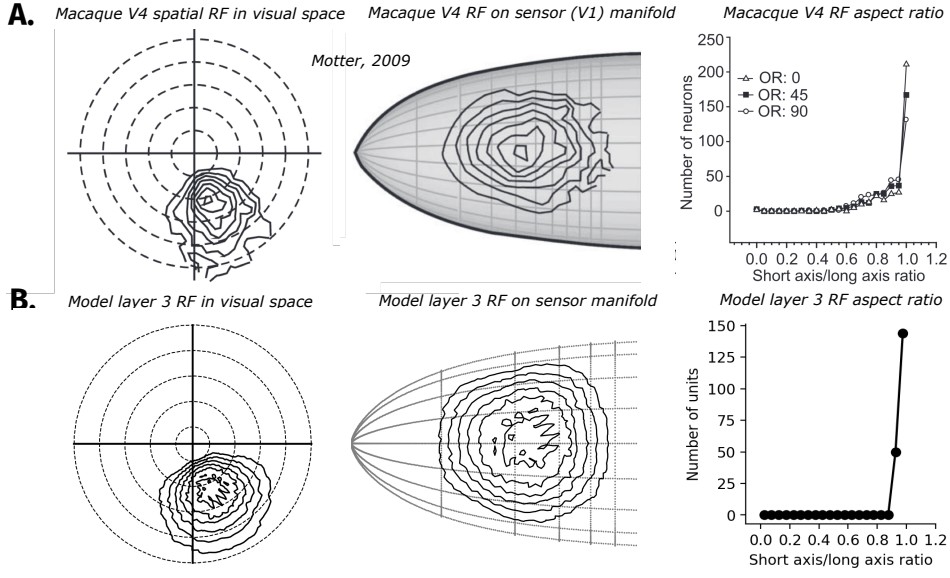

**Figure S3:** kNN-CNN produces biological receptive field properties. **A.** Macaque V4 spatial RF from Motter (2009), plotted in visual (left) and sensor manifold space (middle). On the right, a histogram of aspect ratio is plotted across three model fitting orientations (see Motter (2009) for further detail). **B.** Example aggregated spatial receptive field from a unit in layer 3, plotting in visual space (left) and sensor manifold space (middle). On the right, a histogram of aspect ratio is plotted

## 8.4 PRIOR FOVEATED SENSORS

**Cortical magnification and the complex logarithmic mapping function**   In this section, we discuss in detail prior work developing foveated sensors, and the issues inherent to them which we aim to address in the present work. Such work dates back at least to the pioneering work of Schwartz (1977; 1980), who demonstrated that a logarithmic mapping function could produce accurate fits of cortical magnification, mapping representation across the visual field to representation across primary visual cortex (V1). In key work, Daniel & Whitteridge (1961) defined the cortical magnification factor (CMF) as the amount of cortical space spanned by a fixed amount of retinal (or visual) space, finding that the CMF decreases sharply with eccentricity, but is roughly constant at all points in the visual field of constant eccentricity. This is known as *isotropic* cortical magnification, as the sampling rate is the same at a given point regardless of which direction it is measured. The search for a locally isotropic mapping function well matched to the empirical CMF, which is well fit by a function $M(r) = \frac{1}{r+a}$ (Van Essen et al., 1984, see Figure S1A.), led Schwartz to develop the complex logarithmic mapping model of V1 cortical magnification: $w = \log z + a$, where $z = x + iy$ is the complex plane, and whose derivative is the cortical magnification function (Figure S1E). Along the horizontal meridian, this has the desired CMF $M(r) = \frac{1}{r+a}$, however, while the CMF is locally isotropic, it does exhibit a meridional anisotropy, in which the the CMF along the vertical meridian is maximally different from that along the horizontal meridian, reaching a theoretical maximum ratio of $\sqrt{(2)}$ at $r = a$, at odds with empirical data (Himmelberg et al., 2023).

**The log-polar image model**   As seen in Figure S1, the complex log model leads to a sensor manifold that is curved and disjointed at the vertical meridian, making computer vision applications difficult. A simplified version of this logarithmic mapping approach was thus developed, known as the log-polar mapping. This approach produces a grid-like image output, by simplifying the complex log $\log z + a = \log re^{i\theta} + a$ as two dimensions of $\log r + a$ and $\theta$, where independent sampling can be done along each dimension. However, since an equal number of angles is selected at each radius, the resolution along the angular dimension is highly eccentricity-dependent, and this depends on the value of $a$. If $a = 0$, this approach would be correct, since $\log(re^{i\theta}) = \log(r) + i\theta$, that is, the complex plane of $r$ and $\theta$, matching the approach first developed by (Weiman & Chaikin, 1979). However, the log has a singularity at $a = 0$, and thus cannot be used in practice to model

foveal eccentricities. Setting $a > 0$ removes the foveal singularity, but grid-sampling then introduces anisotropy. As a result, circular receptive fields drawn on a log-polar image with $a > 0$ can have very different shapes across various eccentricities. As $r$ increases, the aspect ratio of receptive fields increases in the tangential direction; for reasonably large values of $a$ (which can be helpful in not oversampling low to medium resolution images), this results in highly elongated peripheral receptive fields, in stark disagreement with empirical data (Motter, 2009). Thus, there is an inherent trade-off: at small values of $a$, the foveal magnification becomes extreme, but anisotropy is less severe, whereas at larger values of $a$, the foveal magnification becomes more realistic, but anisotropy becomes more severe.

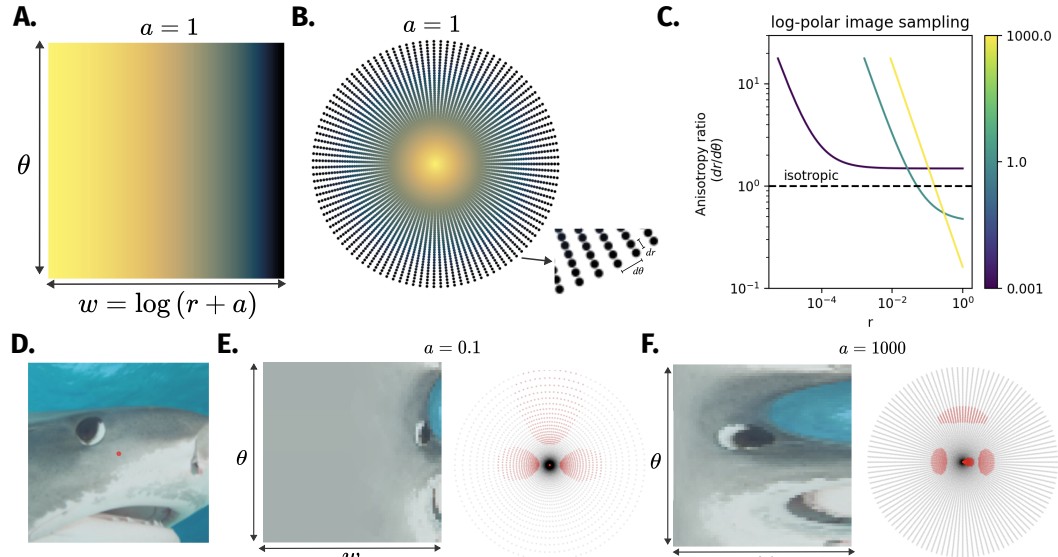

**Figure S4:** The log-polar image approach is anisotropic and introduces oversampling and warped receptive fields. **A.** The log-polar sensor manifold is a regular rectangular (here, square) grid of polar angle and log radius. **B.** The visual sampling induced by equally spaced points in the sensor manifold. Inset: illustration of anisotropy. $dr$ indicates the distance between neighboring radii, while $d\theta$ indicates the distance (arc length) between neighboring angles. **C.** The ratio $dr/d\theta$ is computed locally at each value of $r$ and plotted, for $a \in [0.001, 1, 1000]$. The dashed line corresponds to isotropic sampling. **D.** Illustration of log-polar foveation. Left: a target image with a central fixation point (red dot). Center: log-polar transformed image with $a = 0.1$. Right: log-polar transformed image with $a = 1000$. **E.** Visual receptive fields corresponding to circular samples on the sensor manifold.

**Warped Cartesian approaches** Basu & Licardie (1995) introduced a related foveated sensing approach, also based on the complex logarithm (Schwartz, 1980), that yields a warped Cartesian image. However, as they note, this approach also leads to anisotropies in the periphery. A similar approach was used in recent deep learning approaches by Wang et al. (2021) and Da Costa et al. (2024), which both define a magnification function that depends only on eccentricity, though differing slightly from that inherent to the complex log. However, all of these models can be grouped into the family of warped cartesian image sensors, which show radially elongated receptive fields in the periphery. We illustrate the anisotropy and warping of receptive fields in Figure S5. Additionally, these models do not have a corresponding cortical space that can be mapped to visual cortex, foregoing some of the possible benefits in spatially relating activations in the model directly to brain responses.

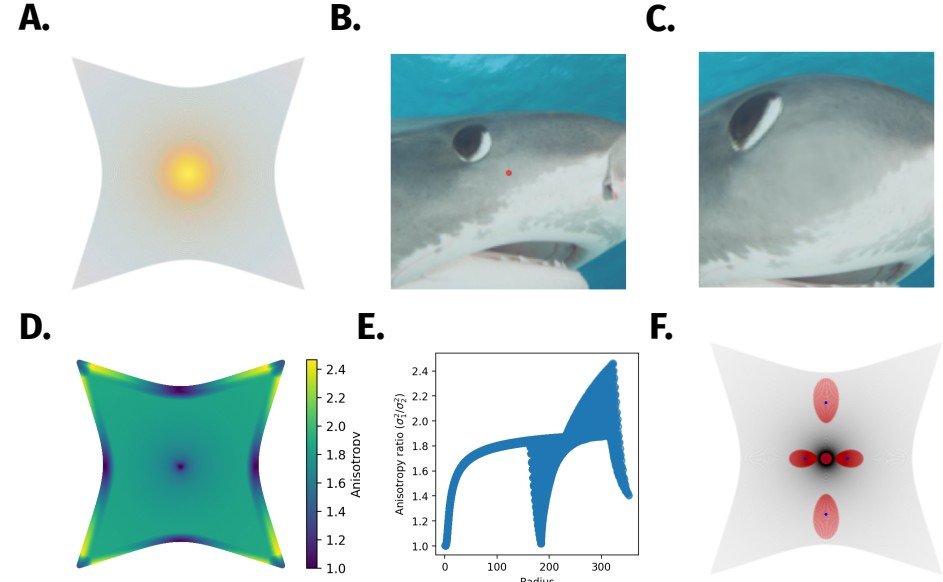

**Figure S5:** Warped Cartesian foveated sensors also introduce anisotropic sampling. Here, we use the sensor described in (Wang et al., 2021). **A**. Sensor locations. **B.** Target image with central fixation point (red circle). **C.** Warped Cartesian foveated image. **D.** Local anisotropy sampling plotted across the visual field. For this approach, at each point in the visual field, we sample $k = 1000$ (of $250^2$) nearest pixel neighbors. Then, we subject the pixel location matrix $X$ to an SVD to find the variance explained by the two principal directions of variance. Given $X^T X = U \Sigma V$, the variance explained by the first and second component are the squared entries of the diagonal matrix $\Sigma$, i.e. $\sigma_1^2$ and $\sigma_2^2$. We then compute the local anisotropy index as $\sigma_1^2 / \sigma_2^2$. **E.** Local anisotropy as computed in **D.**, plotted as a scatter plot against the visual field radius. **F.** Receptive fields drawn as circles in the foveated image space, projected back into visual coordinates.

**Log-Fibonacci sensor** Killick et al. (2023) introduce a foveated sensor that is most similar to ours, in that no attempt is made to wrangle the sensor outputs into an image, producing instead a point-cloud output that is designed to be input to a non-euclidean neural network. However, their model deviates from explicitly modeling cortical magnification in favor of simplicity. Their approach takes advantage of the golden ratio, to achieve sample packing that is approximately uniform within circles in retinal space, as in the seeds of a sunflower. However, while this sensor approach may be a reasonable choice for foveated computer vision, it does not have an associated cortical space, and thus has limited utility in modeling cortical organization, and is more difficult to tune, since it has multiple relevant hyperparameters. Additionally, the authors use structured Gaussian derivative-based filters for processing receptive fields encoded in this sensor space. This is expected to work well in the low to medium data regimes, but in the high data (non-foveated) regime where they were introduced, they were shown to perform worse than standard filters (Jacobsen et al., 2016). This low-data regime is that which was tested by (Killick et al., 2023). Thus, it is unclear how well their approach would scale with increasing data. It would be possible to combine their sensor with our kernel mapping approach, however this is beyond the scope of our paper.

### 8.5 Exploring performance in kNN-CNNs

To better understand our main kNN-CNN results, we performed a series of follow-up analyses. Since it is computationally expensive to run many analyses, we first validate the use of a smaller subset of ImageNet, ImageNet-100, shown in Figure S6B. We find a similar pattern of validation accuracy across foveation levels, with peak performance for the intermediate foveation level of $a = 1$. Here, we additionally assess generalization by comparing accuracy on validation images with that on a matched subset of training images tested after training without data augmentation ("train-match-val"). First, examining ImageNet-1K, we find that more uniformly sampling models tend to overfit the data more, reflected by the increase in train-match-val accuracy without a corresponding increase in validation accuracy; this is seen particularly strongly for a smaller number of fixations. We see the same effect in ImageNet-100, albeit with a greater degree of overfitting. This suggests that foveation allows our models to avoid overfitting on less relevant background information, supporting stronger generalization.

**A.** IN-1K

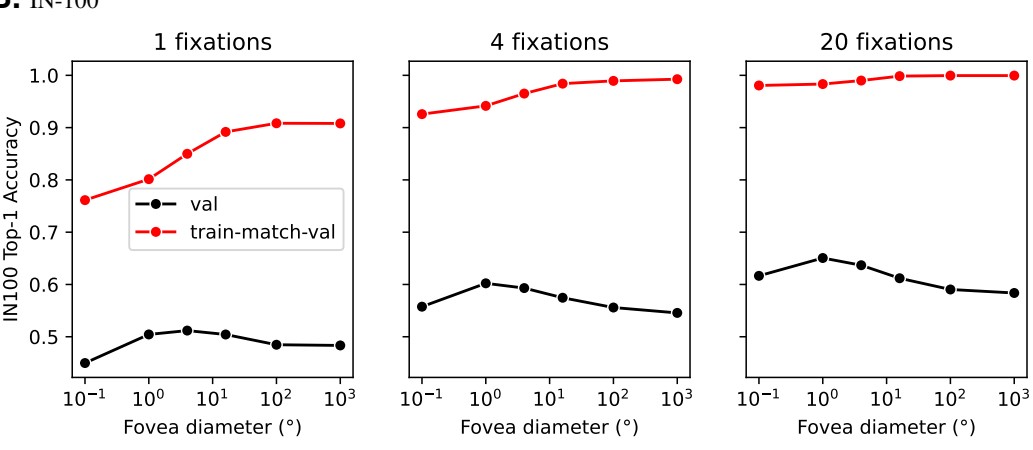

**B.** IN-100

**Figure S6:** Comparing performance for IN-1K and IN-100. For both **A.** and **B.**, we plot performance as a function of the fovea diameter hyperparameter, when evaluated on either validation images, or a matched size sample of training images evaluated without data augmentation. Columns show evaluations using either 1, 4, or 20 fixations, using standard mean-logit aggregation across fixations. **A.** IN-1K results. **B.** IN-100 results.

We next explored the hypothesis that increasing performance for intermediate foveation reflected an **optimal sampling resolution effect**: strong-intermediate foveation works best because it samples near the native resolution in the fovea, neither more (oversampling), nor less (undersampling). One prediction of this account is that decreasing the native resolution of the incoming signal, while

keeping the sampling resolution the same, should shift the accuracy x fovea curves downward (due to less effective resolution), but more importantly, rightward due to a better match of more weakly foveated models (models with larger fovea parameter $a$). Thus, we trained a set of matched models using images that were first resampled at a resolution of 64x64, which we call the native resolution. In Figure S7, we find that both predictions are validated, with performance decreasing overall, but more for more heavily foveated models (i.e., $a = 0.1$ vs. $a = 1000$). However, the advantage for weakly foveated models remains, suggesting that the optimal sampling resolution effect does not fully explain the observed pattern of results.

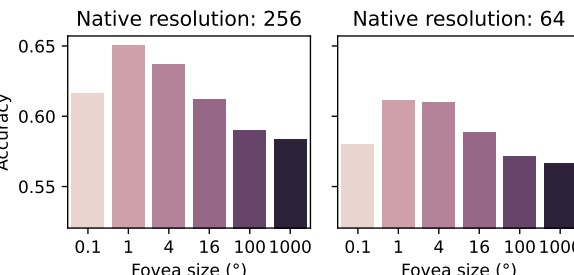 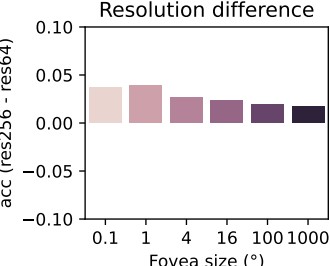

**Figure S7:** Assessing the effect of native image resolution on foveated recognition performance. In the main experiments, we use a native resolution of 256. Here, we plot accuracy using a native resolution of 64 (left) or 256 (middle; standard). On right, we take the difference of the results, for each foveation level.

Our next hypothesis was that the foveation advantage reflects a **central bias of relevant content**, where models with uniform sampling struggle to focus on the central information compared to foveated models. To better understand this, we tested models that incorporate a different form of foveation that could allow for better focus on central information: smaller crops. In our main experiments, we use full-size image crops, but here, we compare with models trained and evaluated using smaller crops of a fraction of 0.2 of the image area; for a 256x256 image, this corresponds to a 114x114 crop. Results are shown in Figure S8. We find that smaller crops lead to worse performance for a single fixation, whereas for the maximum 20 fixations, performance is identical for foveated models, but enhanced for models with more uniform sampling. This suggests that the more uniform models are able to benefit from foveation through cropping, which provides enhanced accuracy over many fixations. However, a large field-of-view provides many real world benefits; here, we capture one of such benefits, which is better performance for a smaller number of fixations.

Last, we explore sampling with a larger fixation zone, using a radius of 0.45 in place of the original radius of 0.25. Interestingly, for large crops, we actually find improved performance for a larger fixation zone, at odds with predictions of the central bias hypothesis.

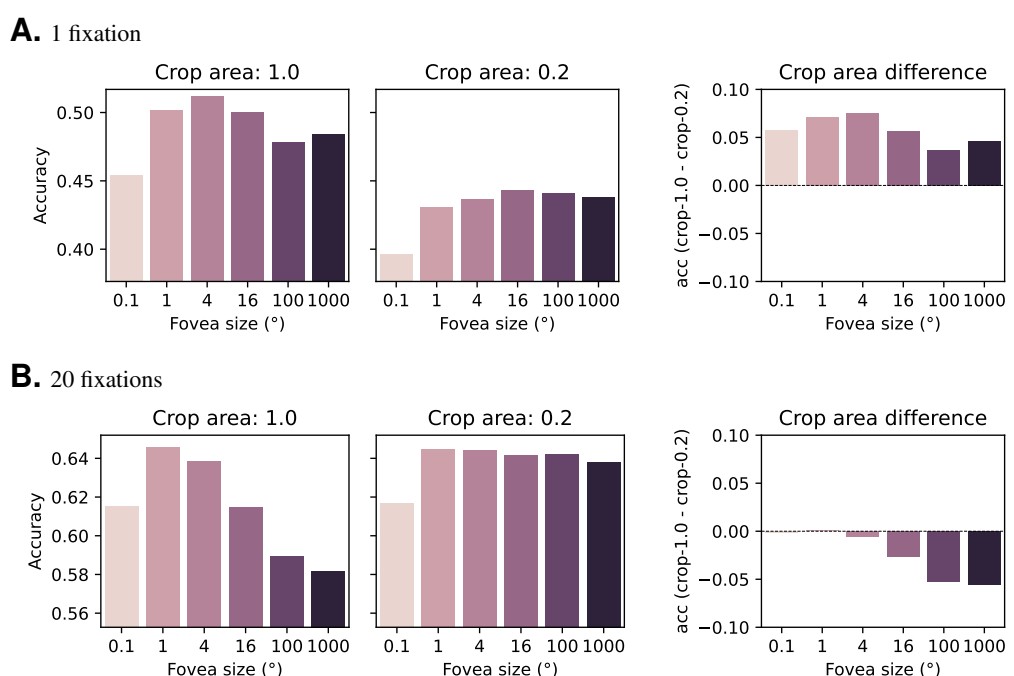

**Figure S8:** Assessing the effect of fixation crop area on foveated recognition performance. In the main experiment, we set the crop area to the full image size. Here, we additionally test a reduced fraction of the image area (0.2). We plot accuracy using the crop area of 0.2 (left) or 1.0 (middle; standard). On right, we take the difference of the results, for each foveation level.

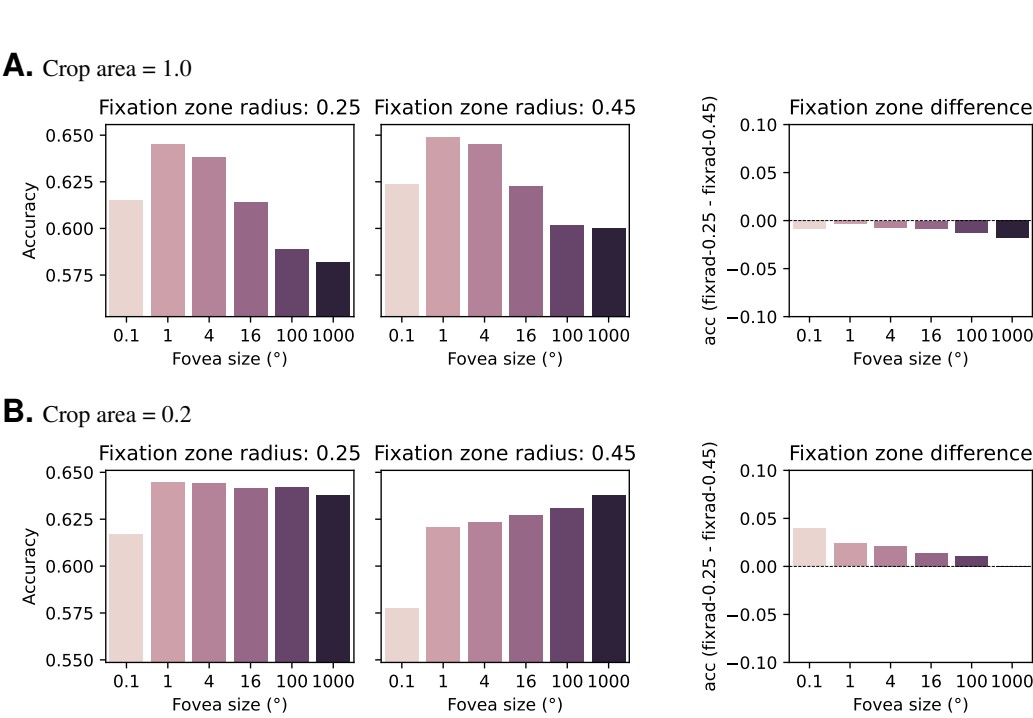

**Figure S9:** Assessing the effect of fixation zone size on foveated recognition performance. In the main experiment, we use a circular fixation zone with a radius of 0.25 of the total image diameter. Here, we compare models trained using a larger fixation zone parameterized by a radius of 0.45. In **A.**, we assess models trained with the standard large crop area (1.0). In **B.**, we assess models trained with the smaller crop area (0.2). In each subpanel, we plot accuracy using the larger fixation zone of 0.45 (left) and standard fixation zone of 0.25 (middle). On right, we take the difference of the results, for each foveation level.

## 8.6 HIGHER RESOLUTION REFERENCE FILTERS IMPROVE KNN-CNN PERFORMANCE

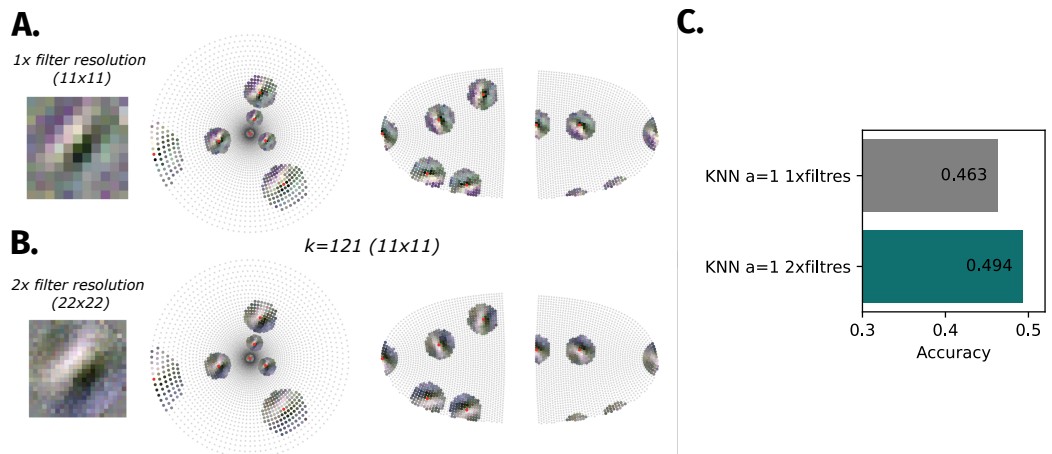

**Figure S10:** Higher reference filter resolution improves performance. **A.** Standard filter resolution for a $k = 121$ kernel (11x11). **B.** Double filter resolution for a $k = 121$ kernel (22x22). **C.** ImageNet-1k performance for foveated kNN-CNN models with 2x and 1x filter resolution, along with a matched CNN.

## 8.7 ADAPTING DINOv3

We explore a variety of methods for adapting DINOv3 to take foveated inputs, comparing to a frozen baseline. Here, to facilitate many experiments, we use the IN-100 dataset as in our hyperparameter explorations of KNN-CNN models. We first note that the frozen baseline performance is significantly reduced relative to the off-the-shelf non-foveated variant operating at a typical 224x224 resolution with 16x16 patch size (93%). Next, we finetune DINOv3 end-to-end, including the foveated patch embeddings. Relative to a frozen backbone, this leads to a significant improvement in validation accuracy, but also a large degree of overfitting (Figure S11A, top). Next, we explore fine-tuning only the first half of the ViT (first 6 layers, indexed as 0-5), along with the patch embedding. This performs similarly, albeit somewhat worse than full fine-tuning. Next, we explore low-rank adaptation (Hu et al., 2021), a method for adapting weight matrices using two low-rank matrices that has been widely successful in preventing overfitting when fine-tuning models on smaller datasets than the original pre-training dataset. LoRA uses the following equation to re-parameterize weight matrices $W$ into low-rank adaptable matrices $\hat{W}$, using two low rank matrices $A$ and $B$:

$$\hat{W} = W + \frac{\alpha}{r} * (BA) \tag{1}$$

Where $\hat{W}$ and $W$ are of shape $(d_{out}, d_{in})$, $A$ is of shape $(r, d_{in})$ and B is of shape $(d_{out}, r)$. We set $r = 8$ and $r = \alpha$ unless otherwise specified. We adapt all weight matrices within a given transformer layer, and explore adapting different combinations of layers.

We find that LoRA over the first half of the network, along with the patch embedding, leads to a significant improvement in validation accuracy (Figure S11A, bottom), along with a large reduction in overfitting. We find that adapting the first half of the network performs best of other strategies we test, including adapting the whole network, only earlier layers, and only the latter half of the network (Figure S11B).

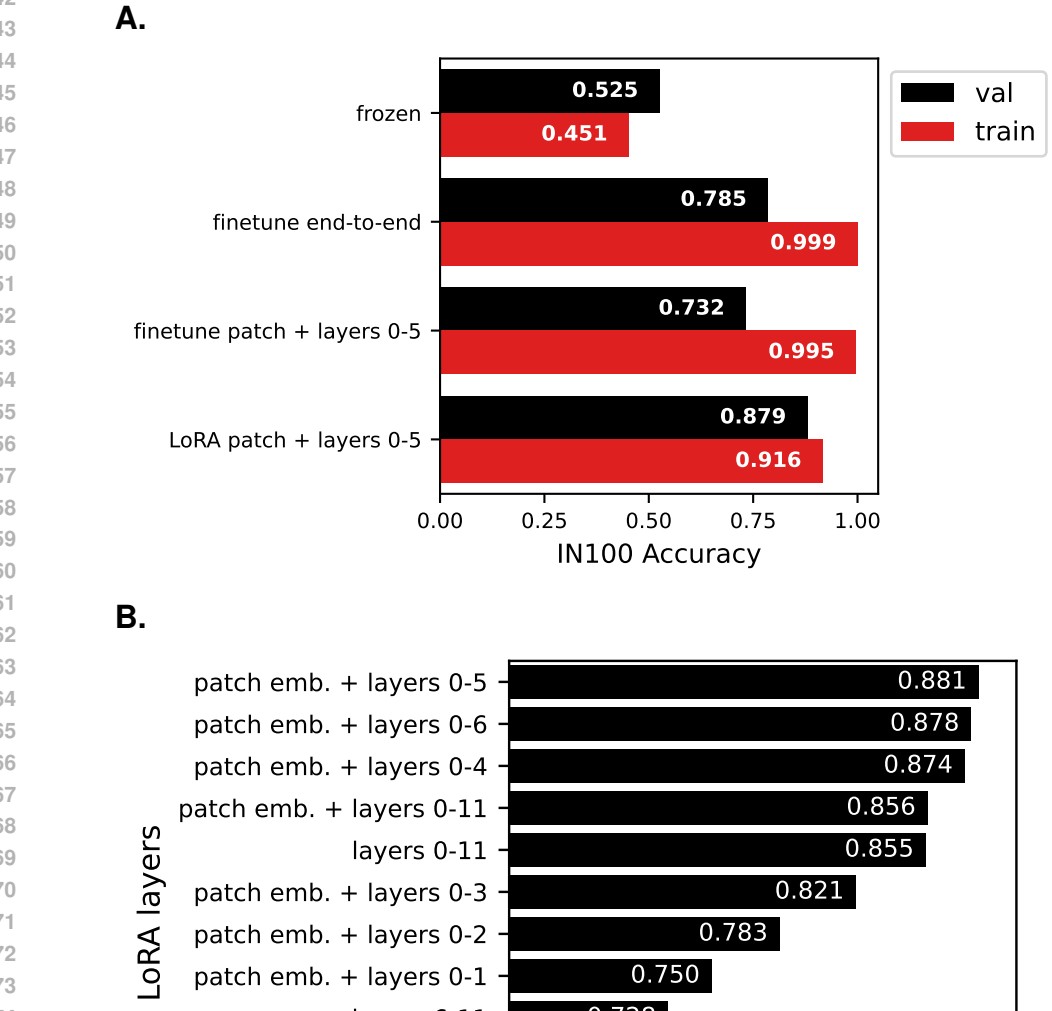

**Figure S11:** Results adapting DINOv3 to process a foveated tokenization ($a = 1$) on ImageNet-100, using different strategies. **A.** Comparing end-to-end finetuning, with early layer finetuning, and early-layer LoRA, for train and val accuracy. Early-layer LoRA training leads to superior generalization and less overfitting. **B.** Comparing a range of LoRA recipes, varying which layers are subject to adaptation. Adapting the patch embedding and first 6 layers produces the best performance in this setting, with significantly worse performance adapting only the first layer or only the late layers.

## 8.8 PERFORMANCE AS A FUNCTION OF FOVEATION DEGREE

Next, we examine the dependence of performance on the foveation parameter $a$. Due to the small number of patches, the inability to exactly determine the number of samples at a given foveation parameter ($a$) can lead to a large percent difference in patch count across models. Thus, rather than choose $a$ directly, here, we constrain the set of $a$ values to those which produce exactly the desired

**A.**                    **B.**

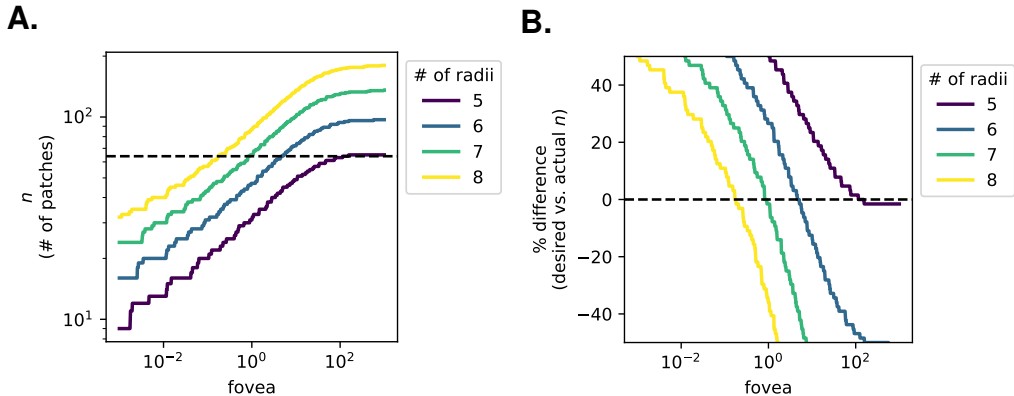

**Figure S12:** Determining a set of fovea values that produce the exact desired number of patches. **A.** Resulting $n$ (here, number of patches) across different combinations of fovea ($a$) and # of sampling radii; note: the # of radii is specified, and the number of sampling points is determined in order to satisfy local isotropy given the particular $a$ value, so it is not fully controllable. **B.** Percent difference in produced $n$ vs. actual $n$. Here, since $n$ is small, the percent difference can be very large. However, by finding the intercepts of each curve, we can specify a set of fovea values that satisfy a perfect match to the desired $n$. Note: we specify a single $a$ per model, so the number of pixels is not exactly matched across models, however the percent difference is much smaller since the desired $n$ is much larger (4096 in our main experiments); for the pixel sampling array, we set the # of radii to the value that most closely matches the target $n$, while not exceeding it.

number of patches. For $n = 64$ patches, we determine 4 suitable $a$ values (rounded here to two decimal places): (0.17, 0.82, 4.61, 115.63), as shown in Figure S12.

We plot the results in Figure S13, using IN-100. We find smaller effects here than with the AlexNet-like CNNs, and a peak performance using $a = 4$ rather than $a = 1$. However, we see a similar inverted U-shaped curve, suggesting again that an intermediate level of foveation is ideal.

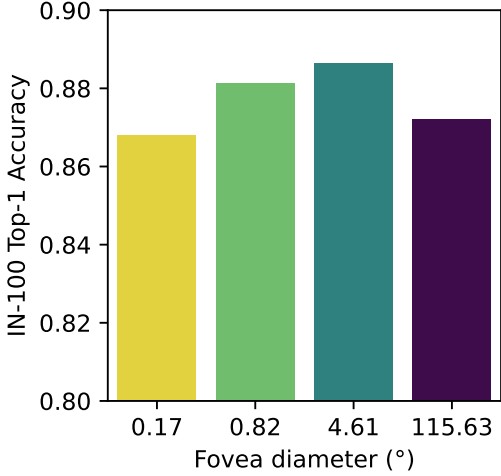

**Figure S13:** Foveated DINOv3 performance on IN-100 as a function of the foveation parameter $a$.

### 8.9 COMPARISON TO LOG-POLAR AND UNIFORM BASELINES

Here, we compare our foveated ViT to matched log-polar and uniform baselines, trained identically, as in the main text. Here, we use IN-100 and explore different fine-tuning strategies for the baseline models, as just done for our model. We find that LoRA improves performance above frozen training (and end-to-end training, for log-polar), so we use the LoRA strategy in the IN-1K-trained models presented in the main text.

**A.**

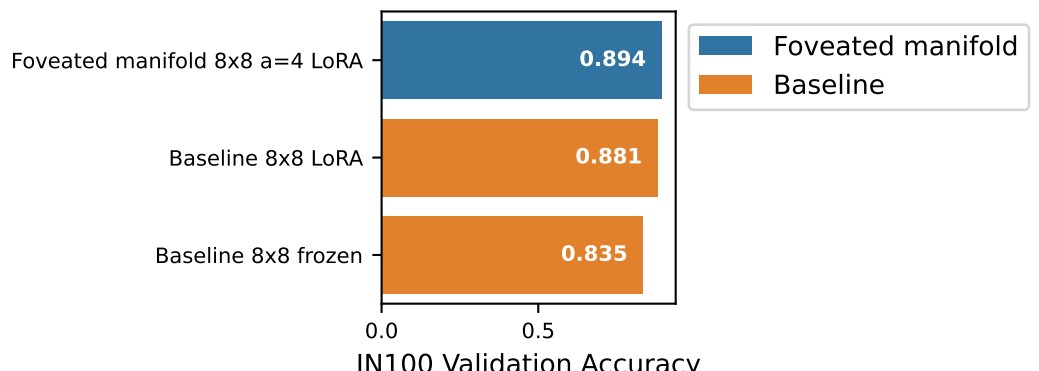

**B.**

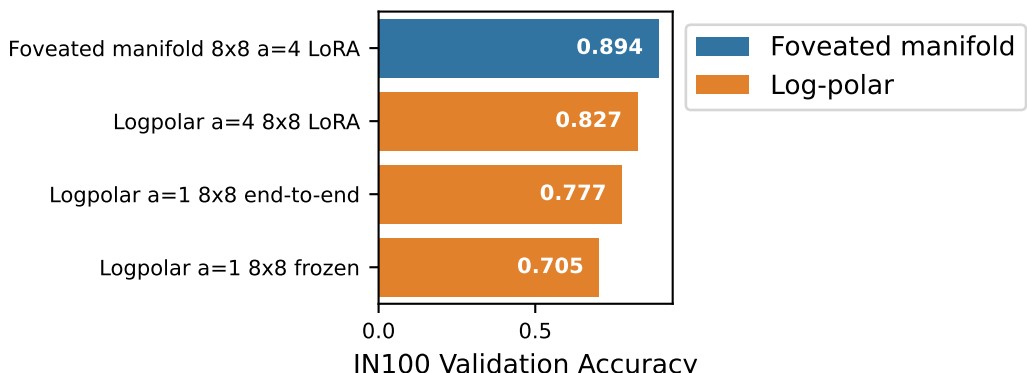

**Figure S14:** Comparing our foveated manifold ViT with a uniform baseline (**A.**) and a log-polar baseline (**B.**), with the ViT adapted with LoRA (patch embedding + first 6 layers, as in the main strategy used for the foveated manifold model), fine-tuned end-to-end, or frozen.

