# OpenReview forum: "A biologically-inspired foveated interface for deep vision models"
_ICLR.cc/2026/Conference — Submitted to ICLR 2026_

### Official Review · Reviewer_zUnm · 2025-10-30

**Soundness:** 2
**Presentation:** 2
**Contribution:** 2
**Rating:** 2
**Confidence:** 4

**Summary:**

The submission implements a foveated approach based on a classic formulation by Rovamo.  They implement a foveated sampling that maps
the visual input to a curved sensor manifold.  They show that the model's receptive fields scale with retinal eccentricity as population receptive fields measured using functional magnetic resonance imaging and compare them to classic Motter single neuron physiology data from monkeys.  The show classification image accuracy with fixation number (random fixations) and show how their foveated approach compares to the non-foveated model.  In the last part of the paper they integrate their approach to a transformer architecture (DINOv3) and show computational savings for the foveation approach.

**Strengths:**

Strengths of the paper are the elegant approach to implementing the foveation.  This seems more involved and different than previous approaches.  The authors implement the approach with CNNs and Transformer approaches.  They evaluate different foveal sizes to evaluate how the model's receptive fields and accuracy vary.  They compare their model to published neuroscience data.

**Weaknesses:**

1.  Why is this approach more valid than previous approaches?
There is no comparison to other foveated approaches that are in the literature:  Freeman & Simoncelli (Nat Neuroscience, 2011).Geisler & Perry, 1988, SPIE).  Why would this one result in better results than others?  Freeman & Simoncelli is fit to neurophysiological data and was used in Akbas et al., PLOS Comp Bio, 2017.
Many of these architectures will result in similar signatures to those described.


2. Problematic results in the paper.

2a. Perhaps the more puzzling result is Fig 3d and e.
It shows that as the fovea gets larger, accuracy increases, but then it gets worse for the non-foveated model (1000 deg fovea diameter)
This is somewhat counterintuitive and needs an explanation.

2b. What do the authors mean by a 1000 deg fovea? The way retinal eccentricity is typically defined, 1000 deg. is not possible.

2c. There is no information about how large the image is assumed to subtend in degrees.  This is

2d. Similarly, it seems strange that ImageNet would improve the classification up to 20 fixations.  Typically, performance will asymptote earlier.



3. The authors focus on getting a foveated approach that can account for neuroscience results but use a random fixation selection which is known not to be optimal or information seeking.  See Najemnik & Geisler, 2005 Nature; Also, Zhou & Eckstein, SPIE, and other papers.

4. The comparison to the population receptive

**Questions:**

Things that could change my mind:

Evidence that their results do not have a mistake or an explanation of their results (see point 2a above)

A more realistic algorithm to generate fixation for the model

Comparison to other foveation approaches.

Strengthen the contribution by emphasizing what is new.

---

### Official Review · Reviewer_UngN · 2025-10-30

**Soundness:** 3
**Presentation:** 3
**Contribution:** 3
**Rating:** 4
**Confidence:** 3

**Summary:**

The paper introduces a biologically inspired foveated sampling interface that maps non-uniform retinal sampling onto a uniformly dense “sensor manifold,” then applies kNN-convolution with a kernel-mapping step so filters preserve orientation/scale across eccentricities. The interface is instantiated in (i) a kNN-CNN (multiple layers) and (ii) a foveated ViT (kNN-conv used only for patch embedding with LoRA). On ImageNet / IN-100, the authors report an inverted-U accuracy vs. foveation curve (best at intermediate foveation) and improvements over uniform downsampling and log-polar baselines at matched pixel/patch budgets.

**Strengths:**

- **Originality:** Clean separation of sensing (how pixels are sampled) from compute (the backbone). The manifold + kernel-mapping design keeps local isotropy and orientation alignment, addressing typical warped-grid/log-polar issues.
- **Quality:** Thoughtful empirical story: sweeping foveation strength reveals a sweet spot (not “more is always better”). The ViT variant is pragmatic, single kNN-conv for embedding, adapted with LoRA.
- **Clarity:** The core construction (manifold, neighborhoods, kernel mapping) and the two instantiations (CNN / ViT) are explained clearly and supported by helpful figures.
- **Significance:** With stronger baselines and a rigorous compute section, the interface is a plausible go-to recipe for high-res perception under tight budgets in vision and embodied settings.

**Weaknesses:**

- **Missing head-to-head with popular foveation families.** Comparisons are mostly to uniform downsampling and log-polar. Readers will expect:
   - *Active hard-attention with learned saccades* (e.g., [1]), which can match strong classifiers while processing only a fraction of the image; and active foveation + learned saccades for localization ([2]) showing gains when the fixation policy is learned (useful motivation to add a small localization task or adapt their policy to classification).
   - *Modern foveated/peripheral Transformers* (e.g., [3], [4]) that inject fovea–periphery inductive bias directly into attention.

   Without at least one active baseline and one foveated-Transformer baseline, matched on tokens/FLOPs and central resolution, it’s hard to isolate the benefit of the proposed interface from generic token/pixel reduction.

- **Compute claims aren’t fully substantiated.** The paper suggests efficiency but does not provide bullet-proof accounting. A fair section should report FLOPs/MACs, wall-clock latency, throughput, peak & activation memory, and ideally energy, under a shared protocol (same GPU, precision, batch, input size). Count all overheads (kNN neighborhood build/lookup, kernel-mapping) and scale by the number of fixations at eval. Disclose any precomputation and whether it’s amortized per sensor layout/parameter $a$.
- **Active-vision gap.** The narrative leans on saccades, yet the method uses random fixations at train/test. A tiny learned policy (uncertainty/top-K heatmap or a lightweight RL head) would make the active-vision story concrete and could shift the accuracy–compute frontier.
- **Ablation depth.** Kernel-mapping details are central. Sensitivity to the reference-kernel resolution, neighborhood size/metric ($k$, geodesic definition), and orientation-alignment robustness should be surfaced in the main text.

---
## References
[1] _Elsayed, G. F., Kornblith, S., Le, Q. V., Ramdas, A., Sohl-Dickstein, J., & Doucet, A. (2019). Saccader: Improving Accuracy of Hard Attention Models for Vision. NeurIPS 2019._
[2] _Ibrayev, T., Nagaraj, M., Mukherjee, A., & Roy, K. (2024, April). Exploring foveation and saccade for improved weakly-supervised localization. In Gaze Meets Machine Learning Workshop (pp. 61-89). PMLR._
[3] _Min, J., Zhao, Y., Luo, C., & Cho, M. (2022). Peripheral Vision Transformer (PerViT). NeurIPS 2022._
[4] _Jonnalagadda, A., Wang, W. Y., Manjunath, B. S., & Eckstein, M. P. (2021). FoveaTer: Foveated Transformer for Image Classification. arXiv:2105.14173._

**Questions:**

1. **Baselines:** Can comparisons to (i) Saccader-style active hard-attention on ImageNet, and (ii) foveated/peripheral Transformer baselines such as Peripheral Vision Transformer and FoveaTer, matched for tokens/FLOPs and central resolution, be added?

2. **Compute accounting:** Including a dedicated section reporting FLOPs, latency (ms/img), throughput, peak & activation memory, and (if feasible) energy (J/img) with the same GPU/precision/batch/image size for all methods would be beneficial. Additionally, clarifying what is precomputed (graph neighborhoods, mapping tables), where it lives (CPU/GPU), and how cost scales with # fixations would also be good.

3. **Fairness:** Alongside uniform/log-polar, including token-reduction / sparse / deformable-attention baselines at matched FLOPs to separate “fewer tokens” from “better foveation” would improve the readability? Also, adding a crop-and-context baseline that preserves high central resolution without foveation would help the readers understand the paper better.

4. **Active policy:** It would be beneficial if the paper could clarify what happens if you replace random fixations with a tiny learned controller (e.g., entropy map with 2–3 steps, or REINFORCE)?

---

### Official Review · Reviewer_f1cH · 2025-10-31

**Soundness:** 2
**Presentation:** 3
**Contribution:** 2
**Rating:** 2
**Confidence:** 4

**Summary:**

This paper addresses the computational challenges of processing high-resolution images in deep learning by proposing a novel, biologically-plausible foveated sampling interface. Inspired by the human visual system's variable resolution (high at the center of gaze, low in the periphery), the authors reformat the foveated inputs into a uniformly dense manifold. To enable processing on this irregular manifold, they introduce a novel method kNN-convolution.

The authors demonstrate two primary use cases:

1. A novel kNN-Convolutional Neural Network (kNN-CNN) architecture that natively learns features over foveated input. This model successfully replicates biological phenomena, such as the increase in receptive field (RF) size with eccentricity, observed in primate visual areas like V1-V3.

2. Integration of the foveated interface into DINOv3, using Low-Rank Adaptation (LoRA).

The results show that the foveated kNN-ViT models maintain or improve accuracy compared to non-foveated, resource-constrained counterparts, demonstrating peak performance at an intermediate level of foveation. The ViT variant, in particular, sets a state-of-the-art for IN-1K at the tested resource constraint. The overall contribution is a general-purpose, biologically-grounded foveation mechanism that opens pathways for efficient and scalable active sensing

**Strengths:**

Overall, this paper is novel and practically useful. The paper is well organized and the experiments can support most of the claims:

* The idea of the converting foveated input to uniformly dense sensor manifold and the kNN-convolution as a general mechanism for biologically-plausible foveated sensing is novel.
* The demonstration that the kNN-CNN natively develops receptive fields matching primate V1-V3 in both size-eccentricity dependence and shape is a strong and unique scientific validation of the interface.
* Through experiments, the foveated ViT variant outperforms non-foveated baselines at the same low pixel and GFLOPs budget, setting a state-of-the-art for this resource-constrained setting on ImageNet-1K.
* The method's integration with DINOv3 using LoRA provides a clear, efficient path for adapting large foundation models, making the approach valuable for practical applications.

**Weaknesses:**

1.Authors mentioned some related previous works and how they are different with the method proposed in this paper. I think it's also good to see the performance comparison under similar budgets with the following works that are also inspired by HVS aiming to use less input budgets to get similar results compare to traditional uniform and dense inputs:

* Liu, J., Bu, Y., Tso, D., & Qiu, Q. (2023, October). Improved efficiency based on learned saccade and continuous scene reconstruction from foveated visual sampling. In The Twelfth International Conference on Learning Representations.

* Lukanov, H., König, P., & Pipa, G. (2021). Biologically inspired deep learning model for efficient foveal-peripheral vision. Frontiers in Computational Neuroscience, 15, 746204.

* Elsayed, G., Kornblith, S., & Le, Q. V. (2019). Saccader: Improving accuracy of hard attention models for vision. Advances in neural information processing systems, 32.

2.The foveated vision normally studies together with attention mechanisms to best take advantage of the input information from limited pixel budget, however in this paper it seems to be ignored. Authors also notice that with more fixation points the model will perform better, and this could be an interesting ablation study. This doesn't necessarily be a weakness, if authors think this is not their focus on this paper and can have an explanation.

**Questions:**

I am happy to increase score if the following questions are answered.

1.In Figure 3D, the CNN will have better performance as more inputs from different fixation points are taken. If my understanding is correct, authors use the average of logits from multiple independent model inferences, each inference is based on input with different fixation points. My question here is I don't see any information fusion as more and more information is received so I do not understand why the more fixation points will help improve the classification accuracy. Each image has the most important feature for classification, when fovea fixation is located on it, the model generally has a better chance of correct prediction vice versa. So according to the author's setting from 5, 10, 15, 20, I don't expect to see any performance improvement.

2.In ViT experiments, the author uses 20 random fixation points, assuming the number of pixels sampled for each fixation point is 6.25%(from figure 3C), a total of 125% pixels are used. Will this contradict to author's claim "reduced computational budget"?

3.In table 1, uniform baseline result is get from downsampling to 64*64 small images to match the same budget with Fov-kNN. In author's previous finding, with increasing fixation points, Fov-kNN should have better performance, but uniform baseline doesn't have this attribute so running it 20 times doesn't make any difference for uniform baseline. My question is in table 1, the data are get from 1 run or 20 runs?

4.Authors select DINOv3 for experiments, since it is not a well cited vit variant may I know why? Intuitively, simply choosing more representative and naive ViT architecture like what the author did for CNNs is more convincing.

---

### Official Review · Reviewer_YB2D · 2025-10-31

**Soundness:** 3
**Presentation:** 3
**Contribution:** 3
**Rating:** 6
**Confidence:** 3

**Summary:**

The paper addresses the computational inefficiency of processing uniformly high-resolution images in deep vision models. It introduces a biologically inspired foveated sampling interface that reformats visual input into a curved, uniformly dense sensor manifold. The method defines receptive fields as k-nearest-neighborhoods and develops a kernel-mapping technique enabling kNN-convolution on this manifold. Experiments with foveated CNNs show receptive-field scaling patterns consistent with primate vision and improved accuracy under pixel-budget constraints. Foveated Vision Transformers adapted from DINOv3 through low-rank adaptation achieve higher ImageNet accuracy than uniform or log-polar baselines at similar computational cost. The results demonstrate efficient visual processing through spatially adaptive sampling while maintaining strong performance.

**Strengths:**

-   The introduction of kNN-convolution is novel and enables flexible spatial processing under foveated sampling.

-   The receptive-field analysis in the Fov-kNN-CNN provides strong biological and mechanistic validation.

-   The Fov-kNN-ViT achieves clear ImageNet-1K gains over matched uniform and log-polar baselines at equal computational cost.

**Weaknesses:**

-   **Modest ImageNet accuracy for Fov-KNN-CNN**
The Fov-kNN-CNN section focuses on biological plausibility; its ImageNet accuracy remains modest and not directly comparable to modern CNNs.

-   **Uniform baseline likely handicapped by global downsampling.** Compressing the full image to 64×64 with 8×8 tokens discards fine detail and departs from pretraining statistics, whereas the foveated variant preserves near-native detail at the center via kNN patchification. This asymmetry can inflate the observed gains at matched FLOPs.

-   **Fixation protocol favors foveation.** Training with 4 fixations and evaluating with 20 benefits the foveated model by repositioning the high-resolution center, while repeated passes over a globally downsampled uniform input provide limited new information. This may bias comparisons in favor of foveation.

- **No evaluation of robustness or invariance.** Foveation is often motivated by efficiency _and_ robustness to visual perturbations. The paper reports only accuracy and GFLOPs. An analysis of adversarial robustness or perturbation stability could strengthen the claim that foveated sampling yields not only efficiency but also perceptual resilience, aligning more closely with biological vision.

- **Limited comparison to other foveation-based approaches.** The paper benchmarks mainly against log-polar and uniform baselines. It omits recent deep foveation models such as **R-Blur** [1] or gaze-contingent methods used in efficient perception. Including such baselines would clarify whether the observed benefits stem from the proposed kNN formulation or from general foveation effects already known in prior work.

1. Training on Foveated Images Improves Robustness to Adversarial Attacks, NeurIPS 2023

**Questions:**

1.  How would the proposed framework extend to multi-object or multi-fixation scenarios where several salient regions must be processed simultaneously? If such capability is absent, does this limit the model’s applicability to real-world scenes with multiple interacting objects?

2.  Given that the method operates on a 2D manifold, would a 3D extension (e.g., depth- or surface-aware foveation) offer advantages in preserving spatial structure and surface continuity?

3.  Are the reported results averaged across multiple random seeds or based on a single run? Reporting variability would strengthen the empirical reliability.


Please also address the weaknesses noted above, particularly regarding the uniform baseline comparison and fixation protocol.

---

### Meta-Review · Area_Chair_Nfvj · 2026-01-12

**Summary:**

- fairness/interpretability of baselines and fixation protocol (e.g., downsampling baseline potentially handicapped; train/eval fixation mismatch)
- insufficient compute accounting (incl. questions about total pixels when using multiple fixations)
- missing comparisons to key related foveation / active-vision / foveated-transformer families, making it hard to isolate whether gains come from the proposed interface vs generic token/pixel reduction

**Reviewer Concerns:**

Addressed:

- No rebuttal was submitted

Still outstanding:

- The uniform baseline is disadvantaged by aggressive global downsampling, and the training/eval fixation setup may inherently favor the foveated method.

- Need rigorous, apples-to-apples reporting (FLOPs, latency, memory, overheads, scaling with #fixations), plus specific concern that 20 fixations could imply >100% pixel usage depending on per-fixation sampling.

- Lack of comparison with active hard-attention / learned saccades and modern foveated transformer baselines (and broader foveation literature), leaving ambiguity about what is truly new/necessary.

- Need explanation for counterintuitive trends and unclear “degree” framing (e.g., “1000 deg fovea”), and why performance continues improving up to many fixations.

- Suggestion to add robustness/invariance analysis and clarify variance across seeds.

**Reviewer Scores:**

As there was no response from the authors, the score would have remained the same:
6, 2, 4, 2

---

### Decision · Program_Chairs · 2026-01-26

Reject